# MoVie: Revisiting Modulated Convolutions for Visual Counting and Beyond

**Duy-Kien Nguyen, Vedanuj Goswami, Xinlei Chen**
Facebook AI Research (FAIR)

## Abstract

This paper focuses on visual counting, which aims to predict the number of occurrences given a natural image and a query (*e.g.* a question or a category). Unlike most prior works that use explicit, symbolic models which can be computationally expensive and limited in generalization, we propose a simple and effective alternative by revisiting modulated convolutions that fuse the query and the image locally. Following the design of residual bottleneck, we call our method *MoVie*, short for *Mo*dulated con*V*olut*i*onal bottl*e*necks. Notably, MoVie reasons implicitly and holistically and only needs a single forward-pass during inference. Nevertheless, MoVie showcases strong performance for counting: 1) advancing the state-of-the-art on counting-specific VQA tasks while being more efficient; 2) outperforming prior-art on difficult benchmarks like COCO for common object counting; 3) helped us secure the *first* place of 2020 VQA challenge when integrated as a module for 'number' related questions in generic VQA models. Finally, we show evidence that modulated convolutions such as MoVie can serve as a general mechanism for reasoning tasks beyond counting.

## 1 Introduction

We focus on visual counting: given a natural image and a query, it aims to predict the correct number of occurrences in the image corresponding to that query. The query is generic, which can be a natural language question (*e.g.* 'how many kids are on the sofa') or a category name (*e.g.* 'car'). Since visual counting requires open-ended query grounding and multiple steps of visual reasoning (Zhang et al., 2018), it is a unique testbed to evaluate a machine's ability to understand multi-modal data.

Mimicking how humans count, most existing counting modules (Trott et al., 2018) adopt an intuition-driven reasoning procedure, which performs counting iteratively by mapping candidate image regions to symbols and count them explicitly based on relationships (Fig. 1, top-left). While interpretable, modeling regions and relations repeatedly can be expensive in computation (Jiang et al., 2020). And more importantly, counting is merely a single visual reasoning task – if we consider the full spectrum of reasoning tasks (*e.g.* logical inference, spatial configuration), it is probably infeasible to manually design specialized modules for every one of them (Fig. 1, bottom-left).

In this paper, we aim to establish a simple and effective alternative for visual counting without explicit, symbolic reasoning. Our work is built on two research frontiers. First, on the synthetic CLEVR dataset (Johnson et al., 2017), it was shown that using queries to directly modulate convolutions can lead to major improvements in the reasoning power of a Convolutional Network (ConvNet) (*e.g.* achieving near-perfect 94% on counting) (Perez et al., 2018). However, it was difficult to transfer this finding to natural images, partially due to the dominance of bottom-up attention features that represent images with regions (Anderson et al., 2018). Interestingly, recent analysis discovered that plain convolutional features can be as powerful as region features (Jiang et al., 2020), which becomes a second step-stone for our approach to compare fairly against region-based counting modules.

Motivated by fusing multi-modalities *locally* for counting, the central idea behind our approach is to revisit convolutions modulated by query representations. Following ResNet (He et al., 2016), we choose bottleneck as our basic building block, with each bottleneck being modulated once. Multiple bottlenecks are stacked together to form our final module. Therefore, we call our method **MoVie**: **Mo**dulated con**V**olut**i**onal bottl**e**necks. Inference for MoVie is performed by a simple, feed-forward pass holistically on the feature map, and reasoning is done implicitly (Fig. 1, top-right).

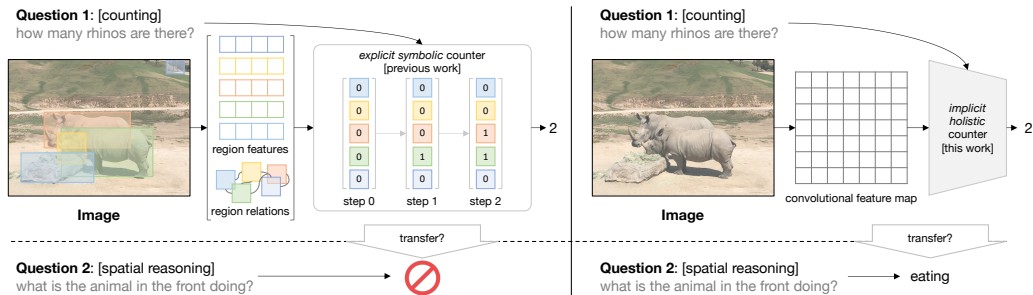

Figure 1: We study visual counting. Different from previous works that perform explicit, symbolic counting (left), we propose an implicit, holistic counter, MoVie, that directly modulates convolutions (right) and can outperform state-of-the-art methods on multiple benchmarks. Its simple design also allows potential generalization beyond counting to other visual reasoning tasks (bottom).

MoVie demonstrates strong performance. First, it improves the state-of-the-art on several VQA-based counting benchmarks (HowMany-QA (Trott et al., 2018) and TallyQA (Acharya et al., 2019)), while being more efficient. It also works well on counting common objects, significantly outperforming all previous approaches on challenging datasets like COCO (Lin et al., 2014). Furthermore, we show MoVie can be easily plugged into generic VQA models and improve the 'number' category on VQA 2.0 (Goyal et al., 2017) – and with the help of MoVie, we won the **first** place of 2020 VQA challenge, achieving 76.36% overall accuracy on the VQA v2.0 test-challenge server.[1] To better understand this implicit model, we present detailed ablative analysis and visualizations of MoVie, and notably find it improves upon its predecessor FiLM (Perez et al., 2018) across all the counting benchmarks we experimented, with a similar computation cost.

Finally, we validate the feasibility of MoVie for reasoning tasks beyond counting (Fig. 1, bottom-right) by its near-perfect accuracy on CLEVR and competitive results on GQA (Hudson & Manning, 2019a). These evidences suggest that modulated convolutions such as MoVie can potentially serve as a general mechanism for visual reasoning. Code will be made available.

## 2 RELATED WORK

Here we discuss related works to the counting module and works related to the task.

**Explicit counting/reasoning modules.** (Trott et al., 2018) was among the first to treat counting differently from other types of questions, and cast the task as a sequential decision making problem optimized by reinforcement learning. A similar argument for distinction was presented in (Zhang et al., 2018), which took a step further by showing their fully-differentiable method can be attached to generic VQA models as a module. However, the idea of modular design for VQA was not new – notably, several seminal works (Andreas et al., 2016; Hu et al., 2017) have described learnable procedures to construct networks for visual reasoning, with reusable modules optimized for particular capabilities (*e.g.* count, compare). Our work differs from such works in philosophy, as they put more emphasis (and likely bias) on interpretation whereas we seek data-driven, general-purpose components for visual reasoning.

**Implicit reasoning modules.** Besides modulated convolutions (Perez et al., 2018; De Vries et al., 2017), another notable work is Relation Network (Santoro et al., 2017), which learns to represent pair-wise relationships between features from different locations through simple MLPs, and showcases super-human performance on CLEVR. The counter from TallyQA (Acharya et al., 2019) followed this idea and built two such networks – one among foreground regions, and one between foreground and background. However, their counter is still based on regions, and neither generalization as a VQA module nor to other counting/reasoning tasks is shown.

Because existing VQA benchmarks like VQA 2.0 also include counting questions, generic VQA models (Fukui et al., 2016; Yu et al., 2019) without explicit counters also fall within the scope of

---

[1]https://visualqa.org/roe.html

'implicit' ones when it comes to counting. However, a key distinction of MoVie is that we fuse multi-modal information *locally*, which will be covered in detail in Sec. 3.1 next.

**Specialized counting.** Counting for specialized objects has a number of practical applications (Marsden et al., 2018), including but are not limited to cell counting (Xie et al., 2018), crowd counting (Sindagi & Patel, 2017), vehicle counting (Onoro-Rubio & López-Sastre, 2016), wild-life counting (Arteta et al., 2016), *etc*. While less general, they are important computer vision applications to respective domains, *e.g.* medical and surveillance. Standard convolution filters are used extensively in state-of-the-art models (Cheng et al., 2019) to produce density maps that approximate the target count number in a local neighborhood. However, such models are designed to deal exclusively with a *single* category of interest, and usually require point-level supervision (Sindagi & Patel, 2017) in addition to the ground-truth overall count number for training.

Another line of work on specialized counting is psychology-inspired (Cutini & Bonato, 2012), which focuses on the phenomenon coined 'subitizing' (Kaufman et al., 1949), that humans and animals can immediately tell the number of salient objects in a scene using holistic cues (Zhang et al., 2015). It is specialized because the number of objects is usually limited to be small (*e.g.* up to 4 (Zhang et al., 2015)).

**General visual counting.** Lifting the restrictions of counting one category or a few items at a time, more general task settings for counting have been introduced. Generalizing to multiple semantic classes and more instances, common object counting (Chattopadhyay et al., 2017) as a task has been explored with a variety of strategies such as detection (Ren et al., 2015), ensembling (Galton, 1907), or segmentation (Laradji et al., 2018). The most recent development in this direction (Cholakkal et al., 2019) also adopts a density map based approach, achieving state-of-the-art with weak, image-level supervision alone. Even more general is the setting of open-ended counting, where the counting target is expressed in natural language questions (Acharya et al., 2019). This allows more advanced 'reasoning' task to be formulated involving objects, attributes, relationships, and more. Our module is designed for these general counting tasks, with the modulation coming from either a question or a class embedding.

## 3 COUNTING WITH MODULATIONS

### 3.1 MODULATED CONVNET

**Motivation.** Why choosing modulated convolutions for counting? Apart from empirical evidence on synthetic dataset (Perez et al., 2018), we believe the fundamental motivation lies in the convolution operation itself. Since ConvNets operate on feature maps with *spatial* dimensions (height and width), the extra modulation – in our case the query representation – is expected to be applied densely to all locations of the map in a fully-convolutional manner. This likely suits visual counting well for at least two reasons. First, counting (like object detection) is a *translation-equivariant* problem: given a fixed-sized local window, the outcome changes as the input location changes. Therefore, a *local* fusion scheme like modulated convolutions is more preferred compared to existing fusion schemes (Fukui et al., 2016; Yu et al., 2017), which are typically applied after visual features are pooled into a single *global* vector. Second, counting requires exhaustive search over all possible locations, which puts the dominating bottom-up attention features (Anderson et al., 2018) that sparsely sample image regions at a disadvantage in *recall*, compared to convolutional features that output responses for each and every location.

**Pipeline and module.** In Fig. 2 (a) we show the overall pipeline. The output convolutional features from a standard ConvNet (*e.g.* ResNet) are fed into the MoVie module at the top.[2] The module consists of four modulated convolutional bottlenecks (Fig. 2 (b)). Each bottleneck receives the query as an extra input to modulate the feature map and outputs another same-sized feature map. The final output after several stages of local fusion is average pooled and fed into a two-layer MLP classifier

---

[2] In fact, the module can be placed in earlier stages, or even be split across different stages. However, we didn't find it help much but incurs extra computation overhead that computes convolutional features *twice* if used as a module for generic VQA models. That's why we stick to the top.

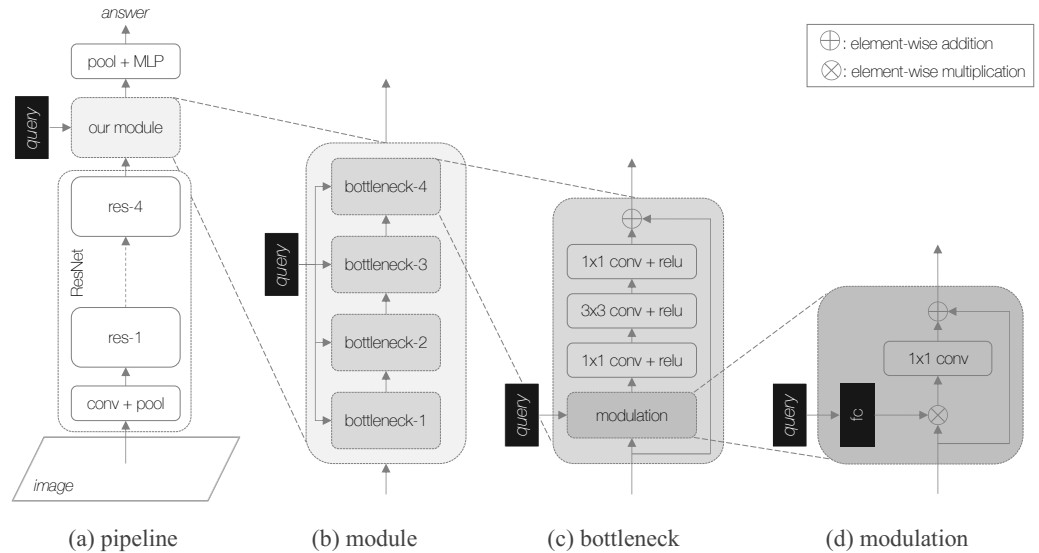

(a) pipeline      (b) module      (c) bottleneck      (d) modulation

Figure 2: **Overview of MoVie** which can be placed on top of any convolutional features. The module consists of several modulated convolutional bottlenecks. Each bottleneck is a simple modified version of the residual bottleneck, with an additional modulation block before the first convolution. The output of the module is average pooled and fed into a two-layer MLP for the final answer.

(with ReLU) to predict the answer. Note that we do *not* apply fusion between query and the global pooled vector: all the interactions between query and image occur in modulated bottlenecks *locally*.

**Bottleneck.** As depicted in Fig. 2 (c), MoVie closely follows the original ResNet bottleneck design, which is both lightweight and effective for learning visual representations (He et al., 2016). The only change is that: before the first $1{\times}1$ convolution layer, we insert a modulation block that takes the query representation as a side information to modulate the feature maps, which we detail next.

**Modulation.** We start with *F*eature-w*i*se *L*inear *M*odulation (FiLM) (Perez et al., 2018) and introduce notations. Since the modulation operation is the same across all feature vectors, we simplify by just focusing on one single vector $\mathbf{v}{\in}\mathbb{R}^C$ on the feature map, where $C$ is the channel size. FiLM modulates $\mathbf{v}$ with linear transformations per channel and the output $\bar{\mathbf{v}}_{\text{FiLM}}$ is:

$$\bar{\mathbf{v}}_{\text{FiLM}} = (\mathbf{v} \otimes \gamma) \oplus \beta, \tag{1}$$

where $\otimes$ is element-wise multiplication, $\oplus$ is element-wise addition (same as normal vector addition). Intuitively, $\gamma{\in}\mathbb{R}^C$ scales the feature vector and $\beta{\in}\mathbb{R}^C$ does shifting. Both $\gamma$ and $\beta$ are conditioned on the query representation $\mathbf{q}{\in}\mathbb{R}^D$ through FC weights $\{\mathbf{W}_\gamma, \mathbf{W}_\beta\}{\in}\mathbb{R}^{D\times C}$.

One crucial detail that stabilizes FiLM training is to predict the *difference* $\Delta\gamma$ rather than $\gamma$ itself, where $\gamma{=}\mathbf{1}\oplus\Delta\gamma$ and $\mathbf{1}$ is an all-one vector. This essentially creates a residual connection, as:

$$\bar{\mathbf{v}}_{\text{FiLM}} = [\mathbf{v} \otimes (\mathbf{1} \oplus \Delta\gamma)] \oplus \beta = \mathbf{v} \oplus [(\mathbf{v} \otimes \Delta\gamma) \oplus \beta].$$

We can then view $\mathcal{F}_{\text{FiLM}} \triangleq (\mathbf{v}\otimes\Delta\gamma)\oplus\beta$ as a residual function for modulation, conditioned jointly on $\mathbf{v}$ and $\mathbf{q}$ and will be added back to $\mathbf{v}$. This perspective creates opportunities for us to explore other, potentially better forms of $\mathcal{F}(\mathbf{v}, \mathbf{q})$ for counting.

The modulation block for MoVie is shown in Fig. 2 (d). The modulation function is defined as: $\mathcal{F}_{\text{MoVie}} \triangleq \mathbf{W}^T(\mathbf{v}\otimes\Delta\gamma)$ where $\mathbf{W}{\in}\mathbb{R}^{C\times C}$ is a learnable weight matrix that can be easily converted to a $1{\times}1$ convolution in the network. Intuitively, instead of using the output of $\mathbf{v}\otimes\Delta\gamma$ directly, this weight matrix learns to output $C$ inner products between $\mathbf{v}$ and $\Delta\gamma$ weighted individually by each column in $\mathbf{W}$. Such increased richness allows the model to potentially capture more intricate relationships. Our final formulation for MoVie is:

$$\bar{\mathbf{v}}_{\text{MoVie}} = \mathbf{v} \oplus \mathbf{W}^T(\mathbf{v} \otimes \Delta\gamma). \tag{2}$$

Note that we also removed $\beta$ and thus saved the need for $\mathbf{W}_\beta$. Compared to FiLM, the parameter count of MoVie can be *fewer*, depending on the relative size of channel $C$ and query dimension $D$.

**Scale robustness.** One concern on using convolutional feature maps directly for counting is on its sensitiveness to input image *scales*, as ConvNets are fundamentally not scale-invariant. Region-based counting models (Trott et al., 2018; Zhang et al., 2018) tend to be more robust to scale changes, as their features are computed on fixed-shaped (*e.g.* 7×7) features regardless of the sizes of the corresponding bounding boxes in pixels (Ren et al., 2015). We find two implementation details helpful to remedy this issue. First is to always keep the input size fixed. Given an image, we resize and pad it to a global maximum size, rather than maximum size within the batch (which can vary based on aspect-ratios of sampled images). Second, we employ multi-scale training, by uniformly sampling the target image size from a pre-defined set (*e.g.* shorter size 800 pixels). Note that input size and image size can be decoupled, and the gap is filled by zero-padding.

**Query representation.** In our experiments, query representations $q \in \mathbb{R}^D$ have two types: questions (Trott et al., 2018) and categories (Chattopadhyay et al., 2017).

For questions, we use LSTM and Self Attention (SA) layers (Vaswani et al., 2017) for question encoding (Yu et al., 2019). Specifically, a question (or sentence in NLP) consisting of $N$ words is first converted into a sequence $\mathbf{Q}^0 = \{w_1^0, \ldots, w_N^0\}$ of $N$ 300-dim GloVe word embeddings (Pennington et al., 2014), which are then fed into a one-directional LSTM followed by a stacked of $L=4$ layers of self attention:

$$\mathbf{Q}^1 = \overrightarrow{\mathrm{LSTM}}(\mathbf{Q}^0), \tag{3}$$

$$\mathbf{Q}^l = \mathrm{SA}_l(\mathbf{Q}^{l-1}), \tag{4}$$

where $\mathbf{Q}^l = \{w_1^l, \ldots, w_N^l\}$, $l \in \{2, \ldots, L+1\}$ is the $D=512$ dimensional embedding for each word in the question after the $(l-1)$-th SA layer.

Given the final $\mathbf{Q}^{L+1}$, to get the conditional vector $\mathbf{q}$, we resort to a summary attention mechanism (Nguyen & Okatani, 2018). A two-layer 512-dim MLP with ReLU non-linearity is applied to compute an attention score $s_n$ for each word representation $w_N^{L+1}$. We normalize all scores by softmax to derive attention weights $\alpha_n$ and then compute an aggregated representation $\mathbf{q}$ via a weighted summation over $\mathbf{Q}^{L+1}$.

For categories, we use an embedding matrix $C_e = \{c_1, ..., c_N\}$ of $N$ category embeddings where $c_n \in \mathbb{R}^D$. The query representation $\mathbf{q}$ of $n$-th category: $\mathbf{q} = \text{L2-Norm}(c_n)$.

## 3.2 MoVie as a Counting Module

An important property of a question-based visual counter is to see if it can be integrated as a *module* for generic VQA models (Zhang et al., 2018). At a high-level, state-of-the-art VQA models (Jiang et al., 2018; Yu et al., 2019) follow a common paradigm: Obtain an image representation $\mathbf{i} \in \mathbb{R}^{C'}$ and a question representation $\mathbf{q} \in \mathbb{R}^D$, then apply a fusion scheme (*e.g.* (Fukui et al., 2016)) to produce the final answer $\mathbf{a}$. A VQA loss $\mathbb{L}$ against ground truth $\hat{\mathbf{a}}$ is computed to train the model (Fig. 3 (a)).

While a naïve approach that directly appends pooled features $\mathbf{v} \in \mathbb{R}^C$ from MoVie to $\mathbf{i}$ (Fig. 3 (b)) likely suffers from feature co-adaptation (Hinton et al., 2012), we follow a three-branch training scheme (Wang et al., 2019; Qi et al., 2020) by adding two *auxiliary*, training-only branches: One trains the original VQA model just with $\mathbf{i}$, and the other trains a normal MoVie with $\mathbf{v}$ and MLP. All three branches are assigned an equal loss weight. The fusion parameters for $\mathbf{i}$ and the joint branch ($\mathbf{i}+\mathbf{v}$) are not shared. This setup forces the network to learn powerful representations within $\mathbf{i}$ and $\mathbf{v}$, as they have to separately minimize the VQA loss. During testing, we only use the joint branch, leading to significant improvements especially on 'number' related questions for VQA *without* sacrificing inference speed (Fig. 3 (c)).

## 4 Experiments

We conduct a series of experiments to validate the effectiveness of MoVie. By default, we use Adam (Kingma & Ba, 2015) optimizer, with batch size 128 and base learning rate $1e^{-4}$; momentum 0.9 and 0.98. We start training by linearly warming up learning rate from $2.5e^{-5}$ for 3 epochs (Yu et al., 2019). The rate is decayed by $0.1\times$ after 10 epochs and we finish training after 13 epochs.

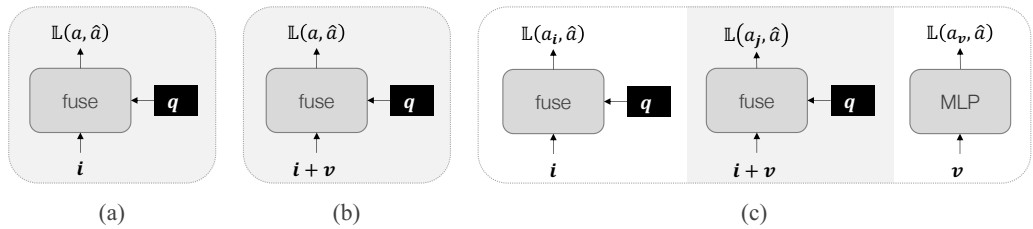

Figure 3: **MoVie as a counting module** for VQA. (a) A high-level overview of the current VQA systems, image **i** and question **q** are fused to predict the answer **a**. (b) A naïve approach to include MoVie as a counting module: directly add pooled features **v** (with one FC to match dimensions) to **i**. (c) Our final design to train with two auxiliary losses on **i** and **v**, while during testing only using the joint branch $\mathbf{a}_j$. Shaded areas are used for both train and test; white areas are train-only.

| Design | $\beta$ | $\oplus$ | ACC ↑ | RMSE ↓ |
|---|---|---|---|---|
| $\bar{\mathbf{v}}_{\text{FiLM}}$ | ✓ | ✓ | 57.1 | 2.67 |
| $\bar{\mathbf{v}}_{\text{MoVie}}$ | ✗ | ✓ | 58.4 | 2.60 |
| $\bar{\mathbf{v}}_{\text{MoVie}}$ variants | ✓ | ✗ | 58.2 | 2.63 |
| | ✗ | ✗ | 57.9 | 2.64 |
| | ✓ | ✓ | 58.5 | 2.63 |

(a) **MoVie** *vs*. FiLM. $\oplus$: residual connection.

| # bottlenecks | ACC ↑ | RMSE ↓ |
|---|---|---|
| 1 | 57.2 | 2.65 |
| 2 | 58.0 | 2.63 |
| 3 | 58.2 | 2.62 |
| 4 | 58.4 | 2.60 |
| 5 | 58.5 | 2.63 |

(b) **Number** of modulated bottlenecks.

| Fixed input size | Test size | ACC ↑ | RMSE ↓ |
|---|---|---|---|
| ✗ | 400 | 22.1 | 4.70 |
| | 600 | 36.0 | 3.16 |
| | 800 | 56.2 | 2.68 |
| ✓ | 400 | 53.9 | 2.92 |
| | 600 | 57.3 | 2.69 |
| | 800 | 58.4 | 2.60 |

(c) **Fixed** *vs*. batch-dependent input size.

| Train size | Test size | ACC ↑ | RMSE ↓ |
|---|---|---|---|
| 800 | 400 | 53.9 | 2.92 |
| | 600 | 57.3 | 2.69 |
| | 800 | 58.4 | 2.60 |
| {400,600,800} | 400 | 56.5 | 2.78 |
| | 600 | 58.8 | 2.66 |
| | 800 | 58.8 | 2.59 |

(d) **Multi-scale** *vs*. single-scale training.

Table 1: **Ablative analysis** on HowMany-QA *val* set. MoVie modulation design outperforms FiLM under fair comparisons; both fixing input size and multi-scale training improve robustness to scales.

## 4.1 VISUAL COUNTING

For visual counting, we have two tasks: open-ended counting with question queries where we ablated our design choices, and counting common object with class queries.

**Open-ended counting benchmarks.** Two datasets are used to counting with question queries. First is HowMany-QA (Trott et al., 2018) where the *train* set questions are extracted from VQA 2.0 *train* and Visual Genome (VG) (Krishna et al., 2017). The *val* and *test* sets are taken from VQA 2.0 *val* set. Each ground-truth answer is a number between 0 to 20. Extending HowMany-QA, the TallyQA (Acharya et al., 2019) dataset augments the *train* set by adding synthetic counting questions automatically generated from COCO annotations. They also split the *test* set into two parts: *test-simple* and *test-complex*, based on whether the question requires advanced reasoning capability. The answers range between 0 and 15. For both datasets, accuracy (ACC, higher-better) and standard RMSE (lower-better) (Acharya et al., 2019) are metrics used for evaluation.

We first conduct ablative analysis on HowMany-QA for important design choices in MoVie. Here, we use fixed ResNet-50 features pre-trained on VG (Jiang et al., 2020), and train MoVie on HowMany-QA *train*, evaluate on *val*. The results are summarized in Tab. 1.

**Modulation design.** In Tab. 1a, we compare MoVie design with FiLM (Perez et al., 2018): All other settings (*e.g.* the number of bottlenecks) are fixed and only $\bar{\mathbf{v}}_{\text{MoVie}}$ is replaced with $\bar{\mathbf{v}}_{\text{FiLM}}$ in our module (see Eq. 1 and 2). We also experimented other variants of $\bar{\mathbf{v}}_{\text{MoVie}}$ by switching on/off

| Method | Backbone | #params (M) | FLOPs (G) | HowMany-QA | | TallyQA-Simple | | TallyQA-Complex | |
|---|---|---|---|---|---|---|---|---|---|
| | | | | ACC ↑ | RMSE ↓ | ACC ↑ | RMSE ↓ | ACC ↑ | RMSE ↓ |
| Count module (2018) | R-101 | 44.6 | - | 54.7 | 2.59 | 70.5 | 1.15 | 50.9 | 1.58 |
| IRLC (2018) | R-101 | 44.6 | - | 56.1 | 2.45 | - | - | - | - |
| TallyQA (2019) | R-101+152 | 104.8 | 1883.5 | 60.3 | 2.35 | 71.8 | 1.13 | 56.2 | **1.43** |
| TallyQA (FG-Only) | R-101 | 44.6 | 1790.9 | - | - | 69.4 | 1.18 | 51.8 | 1.50 |
| MoVie | R-50 | 25.6 | 176.1 | 61.2 | 2.36 | 70.8 | 1.09 | 54.1 | 1.52 |
| MoVie | R-101 | 44.6 | 306.9 | 62.3 | **2.30** | 73.3 | 1.04 | 56.1 | **1.43** |
| MoVie | X-101 | 88.8 | 706.3 | **64.0** | **2.30** | **74.9** | **1.00** | **56.8** | **1.43** |

Table 2: **Open-ended counting** on Howmany-QA and TallyQA *test* set. MoVie outperforms prior arts with lower #parameters and FLOPs. R: ResNet (He et al., 2016); X: ResNeXt (Xie et al., 2017).

$\beta$ and the residual connection ($\oplus$). The results indicate that: 1) The added linear mapping ($\mathbf{W}$) is helpful for counting – all MoVie variants outperform original FiLM; 2) the residual connection also plays a role and benefits accuracy; 3) With $\mathbf{W}$ and $\oplus$, $\beta$ is less essential for counting performance.

**Number of bottlenecks.** We then varied the number of modulated bottlenecks, with results shown in Tab 1b. We find the performance saturates around 4, but stacking multiple bottlenecks is useful compared to using a single one. We observe the same trend with FiLM but with weaker performance.

**Scale robustness.** The last two tables 1c and 1d ablate our strategy to deal with input scale changes. In Tab. 1c we set a single scale (800 pixels shorter side) for training, and vary the scales during testing. If we only pad images to the largest size *within* the batch, scale mismatch drastically degenerate the performance (top 3 rows). By padding the image to the maximum fixed input size, it substantially improves the robustness to scales (bottom 3 rows). Tab. 1d additionally adds multi-scale training, showing it further helps scale robustness. We use both strategies for the rest of the paper.

**Open-ended counting results.** We report *test* set on counting questions for both HowMany-QA and TallyQA in Tab. 2. Even with ResNet-50, MoVie already achieves strong results, *e.g.* outperforming previous work on HowMany-QA in accuracy and TallyQA (FG-Only). With a ResNeXt-101 backbone (Xie et al., 2017), we can surpass all the previous models by a large margin *e.g.* ∼4% in absolute accuracy on HowMany-QA. The same also holds on TallyQA, where we perform better for both simple and complex questions. Note we also have lower parameter counts and FLOPs.

**Visualization.** We visualize the activation maps produced by the last modulated bottleneck for several complex questions in TallyQA (Fig. 4). Specifically, we compute the normalized $\ell_2$-norm map per-location on the feature map (Malinowski et al., 2018). The attention on the question is also visualized by using attention weights (Nguyen & Okatani, 2018) from the question encoder. First two rows give successful examples. On the left, MoVie is able to focus on the relevant portions of the image to produce the correct count number, and can extract key words from the question. We further modified the questions to be more general (right), and even with a larger number to count, MoVie can give the right answers. Four failure cases are shown in the last two rows, where the model either fails to predict the correct answer, or produces wrong attention maps to begin with.

**Common object counting.** Second, we apply MoVie to common object counting, where the query is an object category. Following standard practice, we choose ResNet-50 pre-trained on ImageNet (Deng et al., 2009) as backbone which is also fine-tuned. Due to the skewed distribution between zero and non-zero answers, we perform balanced sampling during training.

Results on COCO (Lin et al., 2014) are summarized in Tab. 3, where we compared against not only the state-of-the-art counting approach CountSeg (Cholakkal et al., 2019), but also the latest improved version of Faster R-CNN. In addition to RMSE, three variants are introduced in (Chattopadhyay et al., 2017) with different focuses. MoVie outperforms all the methods on three metrics and comes as a close second for the remaining one *without* using any instance-level supervision. For more details and results on Pascal VOC (Everingham et al., 2015), see appendix.

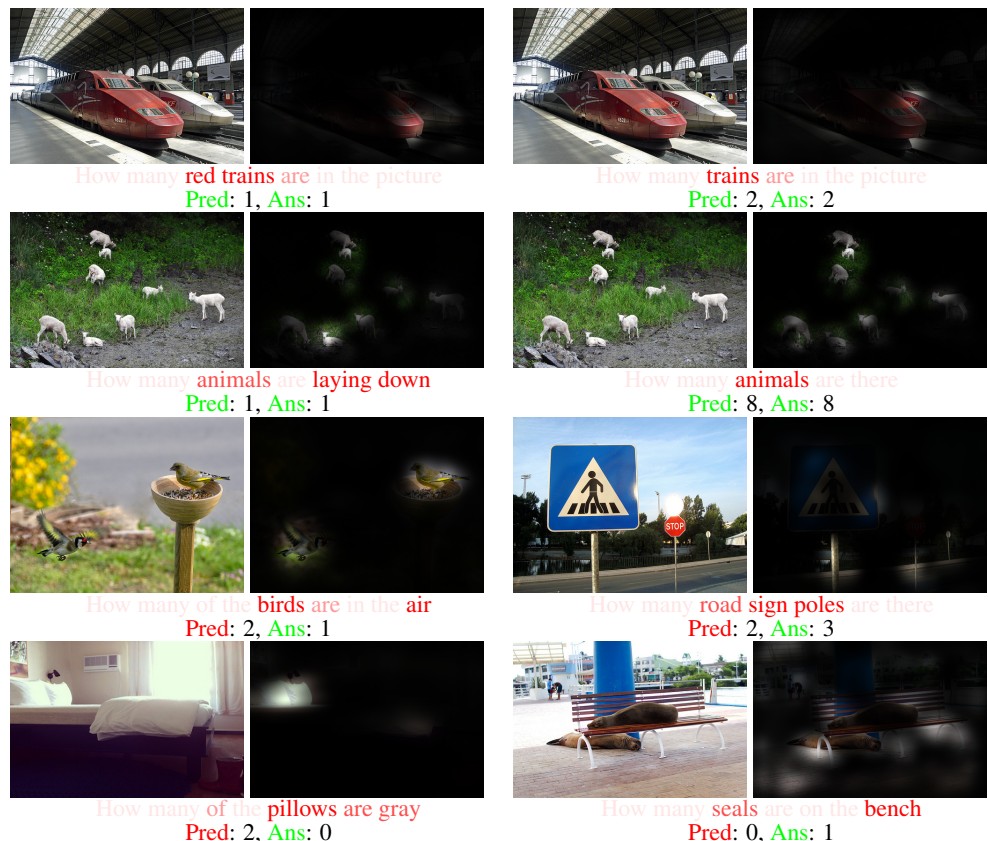

Figure 4: Visualizations of attention maps on images and questions for several complex examples in TallyQA. First two rows show successful cases, and last two shows four failure ones. Best viewed in color on a computer screen. See text for detailed explanations.

| Method | Instance supervision | RMSE ↓ | RMSE-nz ↓ | rel-RMSE ↓ | rel-RMSE-nz ↓ |
|---|---|---|---|---|---|
| LC-ResFCN (2018) | ✓ | 0.38 | 2.20 | 0.19 | 0.99 |
| glance-noft-2L (2017) | ✗ | 0.42 | 2.25 | 0.23 | 0.91 |
| CountSeg (2019) | ✗ | 0.34 | 1.89 | **0.18** | 0.84 |
| Faster R-CNN (2019) | ✓ | 0.35 | 1.88 | **0.18** | 0.80 |
| MoVie | ✗ | **0.30** | **1.49** | 0.19 | **0.67** |

Table 3: **Common object counting** on COCO *test* with various RMSE metrics. MoVie outperforms prior state-of-the-arts without instance-level supervision. All models use ResNet-50 backbone pre-trained on ImageNet.

## 4.2 VISUAL QUESTION ANSWERING

Next, we explore MoVie as a counting module for generic VQA models. We choose MCAN (Yu et al., 2019), the 2019 VQA challenge winner, as our target model (see appendix for Pythia (Jiang et al., 2018), the 2018 winner). For fair comparisons, we only use single-scale training, and adopt the learning rate schedule from MCAN to train MoVie using fixed ResNet-50 features.

The top section of Tab. 4 shows our analysis of different designs to incorporate MoVie into MCAN-Small (Sec. 3.2). We train all models on VQA 2.0 *train* and report the breakdown scores on *val*. Trained individually, MoVie outperforms MCAN in 'number' questions by a decent margin, but lags behind in other questions. Directly adding features from MoVie (naïve fusion) shows limited im-

| Method | Test set | Yes/No ↑ | Number ↑ | Other ↑ | Overall ↑ |
|---|---|---|---|---|---|
| MoVie | | 82.48 | 49.26 | 54.77 | 64.46 |
| MCAN-Small (2019) | *val* | 83.59 | 46.71 | 57.34 | 65.81 |
| MCAN-Small + MoVie (naïve fusion) | | 83.25 | 49.36 | 57.18 | 65.95 |
| MCAN-Small + MoVie | | 84.01 | 50.45 | 57.87 | 66.72 |
| MCAN-Large + X-101 (2020) | *test-dev* | **88.46** | 55.68 | 62.85 | 72.59 |
| MCAN-Large + X-101 + MoVie | | 88.39 | **57.05** | **63.28** | **72.91** |

Table 4: Top: Different ways to use MoVie as a **counting module** for VQA models measured by VQA score (Antol et al., 2015). Bottom: MoVie especially helps 'number' questions on *test-dev*.

| Method | BottomUp (2018) | MAC (2018) | NSM* (2019b) | LCGN (2019) | MoVie | Humans |
|---|---|---|---|---|---|---|
| Overall ↑ | 49.7 | 54.1 | 63.2 | 56.1 | 57.1 | 89.3 |
| Binary ↑ | 66.6 | 71.2 | 78.9 | - | 73.5 | 91.2 |
| Open ↑ | 34.8 | 38.9 | 49.3 | - | 42.7 | 87.4 |

Table 5: **Reasoning** on GQA *test* set to show the generalization of MoVie beyond counting (*: uses scene-graph supervision (Krishna et al., 2017)) Metrics follow (Hudson & Manning, 2019a).

provement, while our final three-branch fusion scheme is much more effective: increasing accuracy on all types with a strong emphasis on 'number' while keeping added cost minimum (see appendix).

In the bottom section of Tab. 4, we further verified the effectiveness of MoVie and the fusion scheme on the *test-dev* split of VQA 2.0, switching the base VQA model to MCAN-Large, the backbone to ResNeXt-101, and using more data (VQA 2.0 *train+val* and VG) for training. We find MoVie consistently strengthens the counting ability. Thanks to MoVie, our entry won the **first** place of 2020 VQA challenge, achieving a *test-std* score of 76.29 and *test-challenge* 76.36. The improvement over Tab. 4 mainly comes from better grid features and model ensembling[3].

### 4.3 BEYOND COUNTING

Finally, to explore the capability of our model beyond counting, we evaluate MoVie on the CLEVR dataset (Johnson et al., 2017). We train our model with a ImageNet pre-trained ResNet-101 for 45 epochs, and report a (near-perfect) *test* set accuracy of 97.42% – similar to the observation made in FiLM. This suggests it is the general idea of modulated convolutions that helped achieving strong performance on CLEVR, rather than specific forms presented in (Perez et al., 2018).

Since CLEVR is synthetic, we also initiate an exploration of MoVie on the recent natural-image reasoning dataset, GQA (Hudson & Manning, 2019a). We use ResNeXt-101 from VG and report competitive results in Tab. 5 *without* using extra supervisions like scene-graph (Krishna et al., 2017). Despite simpler architecture compared to models like MAC (Hudson & Manning, 2018), we demonstrate better overall accuracy with a larger gain on open questions.

## 5 CONCLUSION

In this paper, we propose a simple and effective model named MoVie for visual counting by revisiting modulated convolutions that fuse queries to images locally. Different from previous works that perform explicit, symbolic reasoning, counting is done implicitly and holistically in MoVie and only needs a single forward-pass during inference. We significantly outperform state-of-the-arts on *three* major benchmarks in visual counting, namely HowMany-QA, Tally-QA and COCO; and show that MoVie can be easily incorporated as a module for general VQA models like MCAN to improve accuracy on 'number' related questions on VQA 2.0. The strong performance helped us secure the first place in the 2020 VQA challenge. Finally, we show MoVie can be directly extended to perform well on datasets like CLEVR and GQA, suggesting modulated convolutions as a general mechanism can be useful for other reasoning tasks beyond counting.

---

[3]https://drive.google.com/file/d/1j9QE6xBq7Al_92ylmQEO4Ufq4f5n3Awa/view provides additional details.

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

## A  IMPLEMENTATION DETAILS

We use Pytorch to implement our model on a modular framework for vision and language multi-modal research from Facebook AI Research (FAIR). [4]

**Question representation.** It was shown in Natural Language Processing (NLP) research that adding SA layers helps to produce informative and discriminative language representations (Devlin et al., 2019); and we also empirically observe better results (0.7% improvement in accuracy and 0.1 reduction in RMSE according to our analysis on HowMany-QA (Trott et al., 2018) *val* set). We cap all the questions to the same length $N$ as in common practice and pad all-zero vectors to shorter questions (Nguyen & Okatani, 2018).

Our design for the self-attention layer closely follows (Vaswani et al., 2017) and uses multi-head attentions ($h$=8 heads) with each head having $D_k$=$D/h$ dimensions and attend with a separate set of keys, queries, and values. Layer normalization and feed-forward network are included without position embeddings.

**Object detection based counting.** We train a Faster R-CNN (Ren et al., 2015) with feature pyramid networks (Lin et al., 2017) using the latest implementation on Detectron2 (Wu et al., 2019). For fair comparison, we also use a ResNet-50 backbone (He et al., 2016) pre-trained on ImageNet, the same for our counting module. The detector is trained on the *train2014* split of COCO images, which is referred as the *train* set for common object counting (Chattopadhyay et al., 2017). We train the network for 90K iterations, reducing learning rate by $0.1\times$ at 60K and 80K iterations – starting from a base learning rate of 0.02. The batch size is set to 16. Both left-right flipping and scale augmentation (randomly sampling shorter-side from $\{640, 672, 704, 736, 768, 800\}$) are used. The reference AP (Ren et al., 2015) on COCO *val2017* split is 37.1. We directly convert the testing output of the detector to the per-category counting numbers.

## B  DEFINITION OF RMSE VARIANTS

Besides accuracy and RMSE,[5] object counting (Chattopadhyay et al., 2017) additionally proposed several variants of RMSE to evaluate a system's counting ability. For convenience, we also include them here. The standard RMSE is defined as:

$$\text{RMSE} = \sqrt{\frac{1}{M}\sum_{i=1}^{M}(\hat{c}_i - c_i)^2}, \tag{5}$$

where $\hat{c}_i$ is ground-truth, $c_i$ is prediction, and $N$ is the number of examples. Focusing more on non-zero counts, RMSE-nz tries to evaluate a model's counting ability on harder examples where the answer is at least one:

$$\text{RMSE-nz} = \sqrt{\frac{1}{M_{nz}}\sum_{i\in\{i|\hat{c}_i>0\}}(\hat{c}_i - c_i)^2}, \tag{6}$$

where $M_{nz}$ is the number of examples where ground-truth is non-zero. To penalize the mistakes when the count number is small (as making a mistake of 1 when the ground-truth is 2 is more serious than when the ground-truth is 100), rel-RMSE is proposed as:

$$\text{rel-RMSE} = \sqrt{\frac{1}{M}\sum_{i=1}^{M}\frac{(\hat{c}_i - c_i)^2}{\hat{c}_i + 1}}. \tag{7}$$

And finally, rel-RMSE-nz is used to calculate the relative RMSE for non-zero examples – both challenging and aligned with human perception.

---

[4] https://mmf.sh
[5] https://en.wikipedia.org/wiki/Root-mean-square_deviation

| Method | Instance supervision | RMSE ↓ | RMSE-nz ↓ | rel-RMSE ↓ | rel-RMSE-nz ↓ |
|---|---|---|---|---|---|
| LC-ResFCN (2018) | ✓ | 0.31 | 1.20 | **0.17** | 0.61 |
| LC-PSPNet (2018) | ✓ | 0.35 | 1.32 | 0.20 | 0.70 |
| glance-noft-2L (2017) | ✗ | 0.50 | 1.83 | 0.27 | 0.73 |
| CountSeg (2019) | ✗ | **0.29** | **1.14** | **0.17** | 0.61 |
| MoVie | ✗ | 0.36 | 1.37 | 0.18 | **0.56** |

Table 6: **Common object counting** on VOC *test* set with various RMSE metrics.

| | # Train params (M) | # Test params (M) | Train mem (G) | Train speed (s/iter) |
|---|---|---|---|---|
| MCAN-Large (2019) | 218.2 | 218.2 | 10.5 | 0.84 |
| MCAN-Large + MoVie | 260.2 | 241.2 | 11.7 | 0.89 |

Table 7: Adding MoVie as a module to MCAN. Training speed is ~5% slower, and the additional parameters during testing is minimal (~10%) as it uses the joint branch only.

## C   COMMON OBJECTS COUNTING ON VOC

As mentioned in Sec. 4.1 of the main paper, we present the performance of MoVie on *test* split of Pascal VOC counting dataset in Tab. 6. Different from COCO, the VOC dataset is much smaller with 20 object categories (Everingham et al., 2015). We can see that MoVie achieves comparable results to the state-of-the-art method, CountSeg (Cholakkal et al., 2019) in two relative metrics (rel-RMSE and rel-RMSE-nz) and falls behind in RMSE and RMSE-nz. In contrast, as shown in Tab. 3, MoVie outperforms CountSeg on COCO with a significant margin on three RMSE metrics. The performance difference in two datasets suggests that MoVie scales better than CountSeg in terms of dataset size and number of categories. Moreover, the maintained advantage on relative metrics indicates the output of MoVie is better aligned with human perception (Chattopadhyay et al., 2017).

## D   COUNTING MODULE FOR VQA ARCHITECTURES

As mentioned in 3.2, we integrate MoVie into VQA models as a counting module. Naturally, such an integration leads to changes in model size, training speed *etc*. We report the added computational costs in Tab. 7 for our three-branch fusion scheme into MCAN. We see the training speed is only ~5% slower, and the additional parameters used during testing is kept minimal (~10%). Note that since the integration of MoVie mainly benefits 'number' questions, it is different from general model size increase.

Similar to MCAN, we also conducted experiments incorporating MoVie to Pythia (Jiang et al., 2018), the 2018 VQA challenge winner, where we trained using the VQA 2.0 *train+val*, and evaluated on *test-dev* using the server. We observe even more significant improvements on Pythia for 'number' related questions (Tab. 8). MoVie also improves the performance of the network in all other categories, verifying that our MoVie generalizes to multiple VQA architectures.

## E   VISUALIZATION OF WHERE MOVIE HELPS

When MoVie is used as a counting module for generic VQA tasks, we fuse features pooled from MoVie and the features used by state-of-the-art VQA models (*e.g.* MCAN (Yu et al., 2019)) to jointly predict the answer. Then a natural question arise: where does MoVie help? To answer this question, we want visualize how important MoVie and the original VQA features contribute to the final answer produced by the joint model. We conduct this study for each of the 55 question types listed in the VQA 2.0 dataset (Antol et al., 2015) for better insights.

Specifically, suppose a fused representation in the joint branch is denoted as $\mathbf{o}=f(\mathbf{i}+\mathbf{v})$, where $\mathbf{i}$ is from the VQA model, $\mathbf{v}$ is from MoVie, and $f(\cdot)$ is the function consisting of layers applied after the features are summed up. We can compute two variants of this representation $\mathbf{o}$: one *without* $\mathbf{v}$: $\mathbf{o}_{-\mathbf{v}}=f(\mathbf{i})$, and one *without* $\mathbf{i}$: $\mathbf{o}_{-\mathbf{i}}=f(\mathbf{v})$. The similarity score is then computed between two pairs

| Method | Test set | Yes/No ↑ | Number ↑ | Other ↑ | Overall ↑ |
|---|---|---|---|---|---|
| MCAN-Large + X-101 (2020) | *test-dev* | **88.46** | 55.68 | 62.85 | 72.59 |
| MCAN-Large + X-101 + MoVie | | 88.39 | **57.05** | **63.28** | **72.91** |
| Pythia + X-101 (2018) | *test-dev* | 84.13 | 45.98 | 58.76 | 67.76 |
| Pythia + X-101 + MoVie | | **85.15** | **53.25** | **59.31** | **69.26** |

Table 8: **VQA accuracy** of Pythia with and without MoVie on VQA 2.0 *test-dev* set.

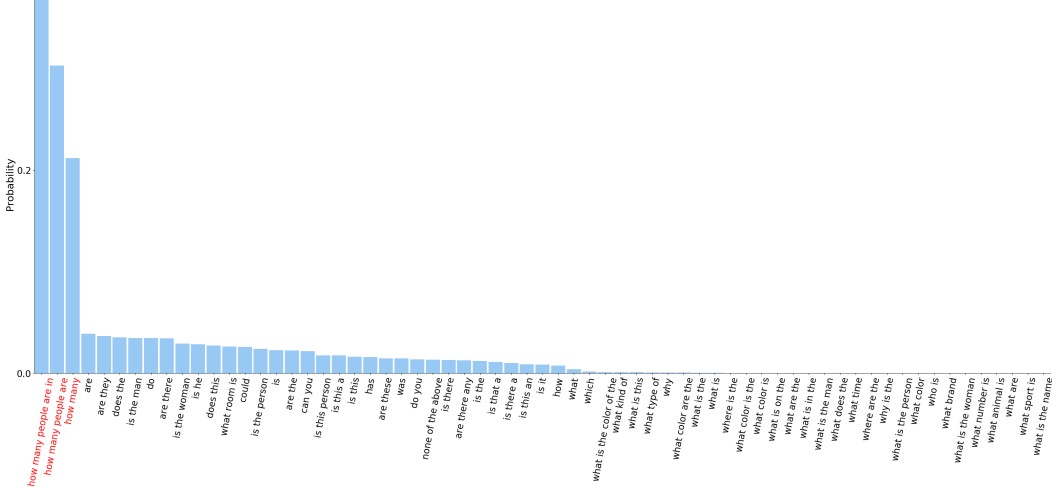

Figure 5: Visualization of where MoVie helps MCAN for different question types on VQA 2.0 *val* set. We compute the probability by assigning each question to MoVie based on similarity scores (see Sec. E for detailed explanations). The top contributed question types are counting related, confirming that state-of-the-art VQA models that perform global fusion are not ideally designed for counting, and the value of MoVie with local fusion. Best viewed on a computer screen with zoom.

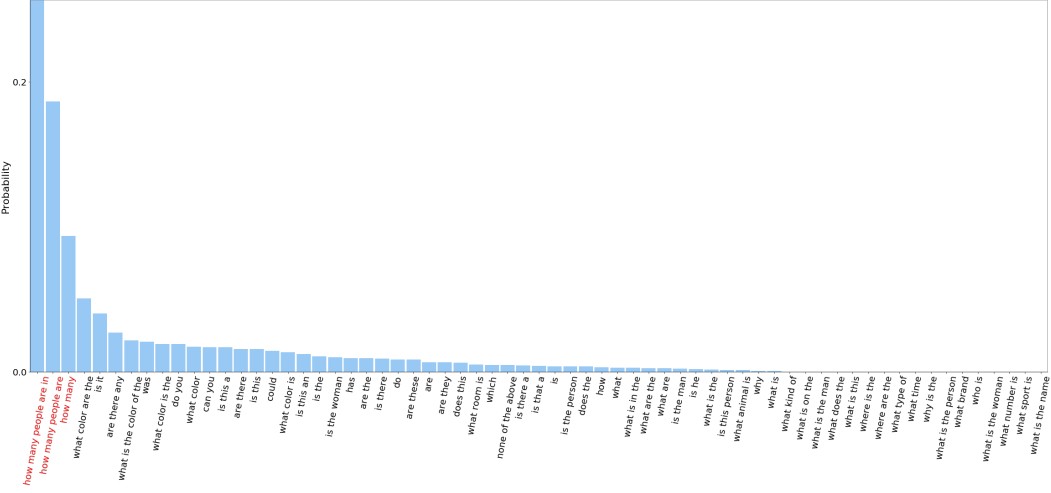

Figure 6: Similar to Fig. 5 but with Pythia. Best viewed on a computer screen with zoom.

via dot-product: $p_\mathbf{i}=\mathbf{o}^T\mathbf{o}_{-\mathbf{v}}$ and $p_\mathbf{v}=\mathbf{o}^T\mathbf{o}_{-\mathbf{i}}$. Given one question, we assign a score of 1 to MoVie if $p_\mathbf{i}>p_\mathbf{v}$, and otherwise 0. The scores within each question type are then averaged, and produces the probability of how MoVie is chosen over the base VQA model for that particular question type.

We take two models as examples. One is MCAN-Small (Yu et al., 2019) + MoVie (three-branch), and the other one replaces MCAN-Small with Pythia (Jiang et al., 2018). The visualizations are

shown in Fig. 5 and Fig. 6, respectively. We sort the question types based on how much MoVie has contributed, *i.e.* the 'probability'. Some observations:

- MoVie shows significant contribution in the counting questions for both MCAN and Pythia: the top three question types are consistently 'how many people are in', 'how many people are', and 'how many', this evidence strongly suggests that existing models that fuse features *globally* between vision and language are not well suited for counting questions, and confirms the value of incorporating MoVie (that performs fusion *locally*) as a counting module for generic VQA models;

- The 'Yes/No' questions are likely benefited from MoVie as well, since the contribution of MoVie spreads in several question types belong to that category (*e.g.* 'are', 'are they', 'do', *etc.*) – this maybe because counting also includes 'verification-of-existing' questions such as 'are there people wearing hats in the image';

- For Pythia, we also find it likely helps 'color' related questions (*e.g.* 'what color are the', 'what color', *etc.*) and some other types – this strengthens our exploration that our model contributes beyond counting capability.

