# OpenReview forum: "MoVie: Revisiting Modulated Convolutions for Visual Counting and Beyond"
_ICLR.cc/2021/Conference — ICLR 2021 Poster_

### Official Review · AnonReviewer2 · 2020-10-27
**A promising approach for counting objects**

**Rating:** 6
**Confidence:** 5

**Review:**

This paper proposes a method to count general objects for a query. Inspired by Perez et al. (2018), the authors design a convolutional neural network consisting of modulated bottlenecks. While the other methods utilize an explicit approach with an object detector, the proposed method is a data-driven and implicit one. Additionally, the authors highlight that MoVie can be integrated with other reasoning tasks. The authors report the experimental results for object counting, visual question answering (VQA), and GQA.

**Pros**
- The proposed approach using modulated convolutions seems reasonable. The motivation is clarified in Sec. 3.1.
- The generalization beyond counting is a good insight for visual reasoning tasks.
- Experimental results show the state-of-the-art performance on counting, VQA, and GQA. Perhaps the authors can compare computational times between the proposed method and the existing methods with symbolic models because MoVie seems faster than them. Table 7 shows the additional computational cost of MoVie, but the reviewer is also interested in comparisons to symbolic models.

**Cons**
- One of the highlighted result is the generalization beyond counting. However, Table 5 only shows the performance with the performance of MoVie in comparison to the other methods. An additional ablation study is necessary to show the contribution of MoVie itself. Table 4 shows Movie boosts the performance not only for the number questions but also for Yes/No questions and the other types for validation data; however, the performance gain is moderate for test-dev data.
- Table 2 shows that the combination of X-101 and MoVie achieves the best performance; as shown, the backbone network is different from the existing methods. The authors should use the same backbone network for a fair comparison.
- The reason why the authors omit beta from Eq. (1) is unclear as far as observed from Table 2. If both beta and residual connections are employed, ACC is slightly better, while RMS is worse than that of MoVie. The significance is also unclear on this table.

**Minor comments**
- Use of "ConvNet" should be avoided since it is confusing whether "ConvNet" is a general convolutional neural network or a particular method or a library.

**Overall rating**
The reviewer is leaning toward acceptance because the motivation is clear and because the proposed method is effective in several experiments. The reviewer would like the authors to solve the cons above to improve the rating.

**Additional comment after rebuttal**
Thanks to to the sufficient answers and results, the first rating is maintained.

---

> ### Author Response · Authors · 2020-11-24
> **] Thanks, we address concerns with extra experiments and updated draft for R2**
>
> Thanks for reviewing our paper and recognizing our motivation, insight, and state-of-the-art performance on several tasks. Below are our responses to the concerns.
>
> **A1** (efficiency comparison to symbolic models): Yes, directly operating on the convolutional feature maps without a symbolic representation based on regions is a major reason that makes MoVie efficient (see Jiang et al for analysis). We did compare MoVie to the TallyQA model (see Table 2) which has symbolic representations for objects, and MoVie can cut the computation to a half or 1/10th while maintaining a similar performance.
>
> **A2** (table 5): Note that in Table 5, we only applied MoVie *by itself* on GQA, and no other state-of-the-art method (like MCAN) is used.
>
> **A3** (table 4): We believe the moderate performance gain on test set is *not* a result of overfitting on val set -- we follow Jiang et al and keep a separate minival set when testing on val. The smaller gain is more likely due to the bigger and more powerful base model (MCAN-Large) used. In fact, as a justification, in Table 8, we show that MoVie can significantly boost the less performant Pythia model by 1.5 percent (67.8 to 69.3) on the same test set.
>
> **A4** (R-101): Thanks for the suggestion. We have run the R-101 backbone with MoVie during the rebuttal period, and updated the draft. Please find the summarization below (ACC is the metric, larger better):
>
> |Method|Backbone|# params (M)|GFLOPs|HowManyQA|TallyQA-Simple|TallyQA-Complex
> |---|---|:---:|:---:|:---:|:---:|:---:
> TallyQA (full)|ResNet-101+152|104.8|1883.5|60.3|71.8|56.2
> TallyQA (no background)|ResNet-101|44.6|1790.9||69.4|51.8
> MoVie|ResNet-101|44.6|306.9|62.3|73.3|56.1
>
> From the table, we can see that MoVie achieves better results than TallyQA while requiring only 1/6 of the computation using the same backbone (mainly because it saves region-level computations).
>
> **A5** ($\beta$): The main reason we remove $\beta$ is for efficiency: the number of parameters in MoVie can thus be comparable to those of FiLM, and not strictly more. As noted by the reviewer, we observe no significant difference with or without $\beta$, so to reduce computation cost and parameter count we choose to not use $\beta$.
>
> **A6** (ConvNet): Thanks for the suggestion, in our context ConvNet stands for general convolutional networks, we have updated the draft to define it when it first appears in text.

---

### Official Review · AnonReviewer1 · 2020-10-27
**Good paper; FiLM to MoVie**

**Rating:** 7
**Confidence:** 3

**Review:**

Description:

This work presents Modulated conVolutional bottlenecks (MoVie) to focus on visual counting where multi-modal information is fused locally. By providing motivation of modulation for visual counting, this paper presents the MoVie module which consists of four modulated convolutional bottlenecks. This paper generalizes FiLM (Perez et al 2018) as a residual function for modulation and introduces a modulation block for MoVie. MoVie can be easily incorporated in any generic VQA on top of any convolutional feature map (eg shown for two models Pythia and MCAN). Extensive ablation studies are presented with SOTA results reported for multiple tasks: open-ended counting with question queries (Howmany-QA and TallyQA) and counting common objects with class queries, apart from the VQA challenge 2.0.

Strength:
- This paper is well-motivated with well-justified compelling results.
- Generalization of FiLM as a residual function for modulation is really interesting and using a learnable weight matrix is intuitive.
- Extensive interesting ablation studies have been provided - number of modulated bottlenecks, modulation design, scale robustness, etc.
- SOTA on multiple counting tasks plus VQA challenge 2.0.
- Extensive analysis has been provided with interesting visualization maps.

Weakness:
- One might argue that anonymity is not properly maintained by mentioning “we won first place in the VQA challenge 2020”. In the future, it is expected to set a precedence by appropriately delexicalizing and stating “we achieve X position in the challenge” (similar to anonymizing Github links)
- In Section 3.2, three branch training is not motivated clearly. The ablation study without three-branch training would help identify the main signal in the training and further strengthen the claims about the generalization of MoVie.
- Implementation details related to the resources, framework, training days, etc. would help in reproducibility apart from the promised code release.

Questions:
- In ML retrospectives Perez et al (2019) noted that the FiLM model could easily overfit the data. Was a similar situation observed with MoVie module? Could the authors further specify how much the results varied for different regularization?
https://ml-retrospectives.github.io/neurips2019/accepted_retrospectives/2019/film/
- In footnote 1, the authors note that the addition of MoVie bottleneck only at the top improved the performance. However, Perez et al. (2019) noted that “Without any neural layers following a FiLM layer, FiLM would have a very limited capacity”. Could the authors comment on this observation?

Suggestions/Comments:
- Please take care of using citep compared to citet (natbib style) appropriately. In the current version, it is really confusing in the main text.
- It would help to bold the best results in Table 1, 4 and 5
- Please be consistent with the usage of “Tab” and “Tables”.
- Please mention the metric used in Table 4 and 5.
- The backbone used for MoVie in Table 3,4 and 5 should be clearly stated.
- Section 4.2 We use choose -> We choose

This is a good paper and it would be interesting to see the discussions to further improve the paper.

--------------------------------------------------------------------------------------------------------------------------------------------------------
Post Rebuttal update:

Thanks to the authors for providing relevant details and fellow reviewers for nice discussions. Original rating is maintained.

---

> ### Author Response · Authors · 2020-11-24
> **Thanks, we address concerns/questions from R1 -- we also incorporated all the suggestions/comments in the updated draft**
>
> Thanks for reviewing our paper and appreciating our effort on motivation, intuition, writing, experimentation, etc. Below are our responses to the concerns.
>
> **A1** (anonymity): Sorry that mentioning VQA challenge rank might have broken anonymity. This is definitely not our intention and we were following previous challenge-winner papers as part of the highlights. We did pay attention to the Github links and did not link it here. We will follow the suggestion next time.
>
> **A2** (three-branch): Intuitively, three-branch training is trying to ensure that features of each branch (MoVie or VQA model) *by itself* can predict answers, so that it squeezes more potential out of them when fused. We will devote more text to explain it clearly in the updated longer version of the paper.
>
> **A3** (implementation details): Yes that’s a good suggestion. We will put the details in the updated version of the appendix. We use PyTorch to implement our model, using a framework called MMF (https://mmf.sh/) (previously known as Pythia).
>
> **A4** (overfitting): Thanks for the reference! We observe that different from FiLM, MoVie doesn't require strong regularization (We don't tune the weight decay and the default value is 0). Its results on both val and test sets are consistent (on both natural and synthetic datasets). Actually, we also observe that MoVie converges faster than FiLM. For example, on CLEVR, with 45-epochs of training, MoVie already reaches 97.4%, whereas FiLM is 2-3% worse.
>
> **A5** (capacity): Several remarks that could potentially explain our observations:
> 1) We hypothesize that, because we are mainly working with natural images (which contains a lot more complexity compared to synthetic images on CLEVR), the *vision-only* process takes much longer before it is semantically meaningful enough to be fused with language input;
> 2) Different from FiLM, MoVie indeed has higher capacity to model the intricate, higher-order relationships between the two modals within the module (note that “L” in “FiLM” stands for “linear”) due to the introduction of $W$, therefore we suspect it reduces the need to have more layers afterwards;
> 3) After all the convolutions and average pooling, we do have an MLP (Figure 2-a) before predicting the answer, so the follow-up neural layer does exist in our pipeline as well.
>
> **A6** (comments): Thanks for the suggestions, we have updated our submission accordingly.

---

### Official Review · AnonReviewer4 · 2020-10-27
**Official Blind Review #4 - post-rebuttal update**

**Rating:** 7
**Confidence:** 4

**Review:**

***************************************************************************
SUMMARY

The paper presents MoVie (short for ‘modulated convolutional bottlenecks’): a new modulation block which reformulates the FiLM approach (Perez et al. 2018) to fuse the query and the image locally, and perform implicit and holistic visual counting. The proposed approach showcases impressive results, establishing new sota both for open-ended and common object counting benchamrks, and competitive results for visual reasoning beyond counting tasks.
***************************************************************************
STRENGTHS

- The paper is fairly well-written, well-structured and easy to follow.
- Obtained results are quite impressive, and establish new sota both for open-ended (Howmany-QA and TallyQA) and common object (COCO) counting benchmarks. The proposed approach also obtains competitive results on VQA (CLEVR and GQA datasets). The authors also say that the proposed approach was a key part of their winning solution of the 2020 VQA challenge.
- The paper also provides a comprehensive ablative study which empirically validates most of the design choices of the approach. This ablative study is completed with qualitative analysis and visualization of some results.
- The paper shows that the proposed approach can be easily plugged into generic VQA systems to improve their ‘counting’ capabilities. This is an interesting property, since, in theory, almost all VQA systems could benefit from this approach.
- The authors plan to make their code public. It could be a valuable resource for researchers working on visual counting (and more generally visual reasoning) and modulated convolutions.
***************************************************************************
WEAKNESSES AND REMARKS

- I have some concerns regarding the novelty of the proposed approach. The paper is revisiting an existing idea (i.e. using modulated convolutions for visual reasoning, which has already been proposed in the FiLM paper) with a new formulation, which gives better results, and thus is applied to natural images (in addition to synthetical ones, like in the FiLM paper).
- The proposed modulation block (a reformulation of the one presented in the FiLM paper) is empirically validated (with a comprehensive and interesting ablative study), but it would have been interesting to give some insights, motivations and theoretical justifications about the key differences compared to the FiLM formulation (i.e. the use of W and the absence of beta - by the way, from Table 1-(a), we can notice that using beta even works slightly better, so I’m not sure to understand the motivations behind this choice-). More globally, the paper contains a lot of empirical choices and lacks theoretical justifications.
- In Table 5, some major recent works on VQA are missing. One can mention, for example, LXMERT (Tan et al., EMNLP 2019) and LCGN (Hu et al., ICCV 2019), both obtaining better results on GQA than those presented in this paper, without being supervised with scene-graphs (unlike NSM).
- It would have been interesting to give more details about the query representation in the paper itself, rather than only referring to the appendix. I understand that it’s due to the lack of space, but it is a little bit frustrating to have no information at all in the paper about this point.
- A similar remark regarding the winning solution of the 2020 VQA challenge. The paper is only mentioning that the proposed approach helped them to secure their first place, without giving any additional information. In my opinion, the paper should either give more details about the winning solution, or keep these information for a dedicated paper without mentioning it in this submission.
- When describing the results presented in Table 1-(b), the paper says that “performance saturates around 4”. But, one can notice that accuracy continues increasing (respectively decreasing for RMSE) when using 5 modulated bottlenecks. It would have been interesting to go further and try using more bottelnecks.
- In my opinion, the ‘beyond counting’ sub-section could be considerably improved if the paper explicitly mention that, unlike CLEVR, the GQA dataset does not contain any counting-related questions. In its present form (without knowing this information), we can assume that the gain in accuracy is simply due to better results on this category of questions, when using MoVie. Knowing that there is no such questions in GQA is crucial to understand that the proposed approach can serve as a general mechanism to improve visual reasoning, beyond counting tasks. It would also have been interesting to evaluate the approach on other visual reasoning tasks beyond VQA (e.g. language-driven comparison of images on the NLVR2 dataset – Suhr et al. ACL 2019- which was used in the FiLM paper) to confirm this claim.
***************************************************************************
JUSTIFICATION OF RATING

Despite the concerns mentioned above, I think the proposed approach is an interesting addition to the visual counting literature, especially considering its quite impressive results. I also think the paper could be considerably improved by taking into account the comments made above. Overall, I'm leaning to accept (6: Marginally above acceptance threshold).

***************************************************************************
POST REBUTTAL UPDATE

The authors provided a detailed rebuttal which addressed almost all my concerns and answered most of my questions. I wil therefore update my rating from 6 (marginally above acceptance threshold) to 7 (good paper, accept). I believe this paper should be accepted.

---

> ### Author Response · Authors · 2020-11-24
> **Thanks, we address concerns for R4**
>
> Thanks for reviewing our paper in detail and listing all the strengths: well-written, strong results, comprehensive study etc. Below are our responses to the concerns.
>
> **A1** (novelty): We focus on making the approach work for real-world images, and the idea of *modulated convolution* is based on our intuition of *local* fusion for counting. While this idea has been explored in FiLM as a specific form before (hence “revisiting”), MoVie is not only a new and better version for natural images per our experiments acknowledged by the reviews, but also a demonstration that *modulated convolution* is the general, key concept that underlies the success of such approaches. Therefore, while revisiting, we believe our work is significant enough for a publication.
>
> **A2** (theoretical justification): We admit that this paper is more about empirical discoveries, and less about theoretical justifications of the design -- and we believe the simplicity of MoVie does reduce the complexity for analyzing the model in the future. Nevertheless, we introduce $W$ mainly to model the intricate, higher-order relationships between the two modals within the module (FiLM standard for *Linear* or *affine* modulation, MoVie is no longer linear), and we remove $\beta$ for efficiency: the number of parameters in MoVie can thus be comparable to those of FiLM, and not strictly more.
>
> **A3** (more comparisons on GQA): Thanks for the references, and we will consider adding those in our Table. However, please note that LXMERT performed vision & language pre-training whereas we did not; and LCGN gets 56.1% on GQA test set compared to 57.1% from MoVie (MoVie is better). Additionally, our goal in this experiment is to show that MoVie can generalize well to other reasoning tasks beyond counting, and given the limited trials (we literally only had a single trial) and the good performance, we are confident to believe that it is indeed the case.
>
> **A4** (query details): Thanks, indeed the query details are removed in the main text due to page limit (and sorry for the inconvenience). We plan to include them in the updated longer version of the paper.
>
> **A5** (challenge details): Thanks, we omitted those details as we do not intend to defocus the theme of the paper (counting). We will give more details about the challenge in the updated longer version of the paper.
>
> **A6** (number of bottlenecks): In Table 1-b, we do find accuracy goes up with 5 bottlenecks, but the RMSE *increases*, note that for RMSE, the smaller the value, the better. So it is saturated.
>
> **A7** (beyond counting): Thanks a lot for the suggestion and we updated the draft to explicitly mention that GQA does not contain counting related questions (we believe CLEVR still does). For NLVR2, as it uses a pair of images as input (whereas we assume a single image input in MoVie), we will leave it for future work.

---

### Official Review · AnonReviewer3 · 2020-11-01
**VQA - object counting, room for improvement**

**Rating:** 6
**Confidence:** 3

**Review:**

Summary:

The paper addresses the counting-based VQA scenarios, as well as general object counting problems. On the VQA side, following [1], they rely on grid-based features/predictions, vs region-based annotation/prediction. Then following [2, 3], they employ LSTM with a self attention layer for the question encoding. As part of the contribution of the paper, they modified FiLM (Feature-wise Linear Modulation) formulation to form an alternative modulation for the task of counting, which they call it MoVie. Here is how the paper reformulate FiLM:
$$\bar{v}_{FiLM}=\left [ v\otimes \left ( 1\oplus\Delta \gamma  \right ) \right ]\oplus \beta =v\oplus\left [ (v\otimes\Delta \gamma)\oplus\beta \right ]$$
, where $v$ represents feature vector, $\gamma$ and $\beta$ are both conditioned on query representation, and $\left ( v \otimes \Delta \gamma\right )\oplus \beta$ as the residual in FiLM formulation.

Differently to FiLM residual function, in MoVie, they added a weight matrix, $W$, to learn $C$ inner products between $v$ and $\delta \gamma$ weighted individually by each column in $W$, to capture more information. In addition, they removed the residual $\beta$. $\bar{v}_{MoVie}=v \oplus W^{T}(v \otimes \Delta \gamma)$. Empirically they showed the presence of $\beta$ is not essential in MoVie formulation.

By removing $\beta$ from the formula, depending on the dimensions of the $W_{\beta}$, and $W$, MoVie could be more or less efficient than the FiLM original formulation.

S: strength W: weakness/ room for improvement

S1: It is intuitive but not trivial that the modification in the FiLM formulation may improve the performance in VQA object counting-related tasks. So the empirical experiments were comprehensive to confirm the assumption, and it is a valuable founding.

S2. W1: They showed MoVie’s performance beyond counting tasks and on more general VQA datasets such as GQA and CLEVR. However, I would like to see how replacing FiLM with MoVie changes the performance.

W2: It is valuable to see the visualization of some failure and successful cases of the model (Figure 4). However, in order to give a better insight on the MoVie advantages vs FiLM, I suggest presenting some cases that FiLM fails but MoVie is able to respond to the query/count correctly.

W3. In the paper, in multiple places it is stated that the approach is more efficient compared to alternative approaches. However it is not really clear to me how it is considered as more efficient. For example, in Table 2, the number of parameters for MoVie is relatively higher than other approaches. Could you elaborate on the efficiency of the model?

W4. For the general counting task, it is presented that MoVie outperforms other methods in the object counting literature on the COCO dataset. However in appendix, it is presented that the model doesn’t have competitive results on the PASCAL VOC dataset not only compared to [4] but also to some other papers in literature. The reason that is mentioned in the paper is related to the scale of the dataset [5]. Object counting datasets are usually compared based on their performance while objects are heavily occluded or in crowd counting tasks. Since the presented failure tasks in Figure 4, are both examples of occlusions, I am wondering how the model performs on popular and challenging crowd counting datasets, such as Penguins, Trancos, or Shanghai datasets.

W5. Image-level object counting, implicit counting, and multimodal reasoning aspects of the tasks were all presented in other papers in the literature (and well-cited within the paper). It is not clear to me besides improving the FiLM formula, and employing it for VQA counting-related and counting tasks, what are the other contributions of the paper?

References:

[1] Jiang et al, In Defense of Grid Features for Visual Question Answering, CVPR 2020.

[2] Vaswani et al, Attention is all you need, NeurIPS 2017.

[3] Yu et al, Deep modular co-attention networks for visual question answering, CVPR 2019.

[4] Cholakkal et al, Object counting and instance segmentation with image-level supervision, CVPR 2019.

[5] Laradji et al, Where are the blobs: Counting by localization with point supervision, ECCV 2018.

######################################

Reason for the decision:
Mentioned points marked as S and W. Adding more clarity to the contributions specific to the paper and comparisons as mentioned,  could change my score.

#####################################
After rebuttal: (from 5 to 6)

Thanks for the clarifying the points in your response. I am happy to change my score to Accept the paper as most my concerns are addressed and I believe the paper is a good fit in this conference.

A comment to the authors that didn't impact my score but raised some concerns:

My concern is about mentioning the first place in VQA challenge 2020, both in paper and also in rebuttal comments. The review process is double blind and pointing out to other contributions that are public and reviewers may have already known about the winner teams, may not be fair. I know that papers can be online on arXiv but pointing to another venue as part of the contributions, may reveal the identity of the authors explicitly (if reviewers already know about the challenge)

---

> ### Author Response · Authors · 2020-11-24
> **Thanks, we address concerns with extra experiments for R3**
>
> Thanks for reviewing our paper and acknowledging the value of our work & the comprehensiveness of our experiments. Below are our responses to the concerns.
>
> **A1**: We only had very limited trials for CLEVR/GQA as the focus of the paper is on visual counting. Nevertheless, as we reported in the paper, with just 45-epochs of training, MoVie reaches 97.4% (FiLM paper trained with 80-epochs). In the same setting, the FiLM module is 2-3% worse per our experiments. We also performed one trial of FiLM on GQA after we saw this request, which also yields worse results than MoVie especially for open-query questions. Further exploration is admittedly needed (for future work), and in this paper we focus on counting.
>
> **A2**: Thanks for the suggestion. We will consider adding side-by-side examples comparing MoVie and FiLM in the updated longer version of the paper.
>
> **A3**: The model is efficient mainly because it directly operates on convolutional feature maps and avoids the extra step of modeling regions -- this is studied in detail in Jiang et al (e.g. 40x speed up with ResNet-50 backbone). It is also reflected in the FLOPs counts in Table 2 (comparing TallyQA vs. MoVie). Note that parameter counts for convolution operators are *independent of input resolution* and hence do not fully reflect the actual capacity or runtime of a convolutional network, therefore are less reflective of the actual efficiency compared to FLOPs.
>
> **A4**:
> 1) We will be more careful with our wording that might have caused confusions, but it is not our intention to overclaim that we outperform state-of-the-art on all object counting benchmarks.
> 2) Although the reason for different behaviors on COCO and PASCAL are hypothesized, we believe we did achieve competitive results on VOC as the top two rows “LC-ResFCN” and “LC-PSPNet” use additional instance supervision for their methods to work well, and MoVie is the second best among the bottom three methods compared, and the best for the metric  “rel-RMSE-nz”.
> 3) Note that crowd-counting does not require modulation at all (as the task does not need query embeddings to specify what to count) so it is beyond the scope of this paper, but we believe our work can bridge the gap between crowd-counting and open-ended counting, because MoVie and popular density-map based methods for crowd-counting are both relying on convolutions (and not separate regions).
>
> **A5**: We believe our contributions are threefold:
> 1) We are the first to show that modulated convolutions can work for real-world, open-ended counting datasets beyond synthetic datasets like CLEVR, by demonstrating strong empirical results on multiple tasks and multiple datasets;
> 2) While we revisit modulated convolutions, our new MoVie module works better than original FiLM across datasets;
> 3) We also propose a new, simple fusion scheme to integrate MoVie (or similar future modules) into any generic VQA models, which works really well and helps us secure the first place in VQA challenge 2020 -- MoVie + MCAN is the state-of-the-art.

---

### Decision · Program_Chairs · 2021-01-07
**Final Decision**

**Decision:**

Accept (Poster)

**Comment:**

The paper proposes a modification of the well-known FILM model for VQA which targets counting problems in particular, which have been a known weakness of existing models. The improvements have also been tested beyond counting. The experimental results are convincing, in particular a scientific competition has been won. The reviewers also appreciated convincing ablation studies.

The idea bas been perceived as interesting enough for publication, and in combination with the experimental results, this compensated several perceived weaknesses (limited novelty w.r.t. the modified FILM model; justifications of some design choices).

All reviewers agreed that this paper is of interest to the community and proposed acceptance. The AC concurs.